# Triple Primary Malignancies: Tumor Associations, Survival, and Clinicopathological Analysis: A 25-Year Single-Institution Experience

**DOI:** 10.3390/healthcare11050738

**Published:** 2023-03-02

**Authors:** Iulia Almasan, Doina Piciu

**Affiliations:** 1Department of Endocrine Tumors and Nuclear Medicine, Institute of Oncology, 400015 Cluj-Napoca, Romania; 2Ph.D. School of Iuliu Hațieganu, University of Medicine and Pharmacy, 400347 Cluj-Napoca, Romania

**Keywords:** synchronous tumors, metachronous tumors, triple primary malignancies, tumor association patterns

## Abstract

The detection of multiple primary malignancies is on the rise despite their rare occurrence rate. This research aims to determine the prevalence, tumor association patterns, overall survival, and the correlation between survival time and independent factors in patients with triple primary malignancies. This single-center retrospective study included 117 patients with triple primary malignancies admitted to a tertiary cancer center between 1996 and 2021. The observed prevalence was 0.082%. The majority of patients (73%) were over the age of fifty at the first tumor diagnosis, and regardless of gender, the lowest median age occurred in the metachronous group. The most common tumor associations were found between genital–skin–breast, skin–skin–skin, digestive–genital–breast, and genital–breast–lung cancer. The male gender and being over the age of fifty at the first tumor diagnosis are associated with a higher risk of mortality. Compared with the metachronous group, patients with three synchronous tumors demonstrate a risk of mortality 6.5 times higher, whereas patients with one metachronous and two synchronous tumors demonstrate a risk of mortality three times higher. The likelihood of subsequent malignancies should always be considered throughout cancer patients’ short- and long-term surveillance to ensure prompt tumor diagnosis and treatment.

## 1. Introduction

Multiple primary malignancies (MPMs) are defined as two or more malignant tumors that are histologically distinct and arise in the same or a different organ. Identifying multiple primary tumors in the same individual is not a recent construct. In 1889, Theodor Billroth [1] described the first case of MPMs, whereas later, in 1932, Warren and Gates [2] established the defining criteria for MPMs as follows: (a) Each primary tumor had to be histologically proven as malignant; (b) each primary tumor had to be anatomically distinct from one another; and (c) it had to be confirmed that one tumor was not a metastatic lesion or recurrence of the other. The first identified malignancy is referred to as the index tumor, while the subsequently diagnosed malignancies are defined as either synchronous (diagnosed within a 6-month time frame) or metachronous (occurring more than six months after the previous malignancy).

According to Global Cancer Statistics 2020, GLOBOCAN, the global cancer burden is estimated to have risen to 19.3 million new cases and is expected to reach 28.4 million by 2040. At the moment, female breast cancer is the most commonly diagnosed type of cancer (11.7%), followed by lung (11.4%), colorectal (10.0%), prostate (7.3%), and stomach cancer (5.6%) [3]. The detection of MPMs is on the rise mainly due to a significant upsurge in new cancer cases, increased longevity, and improved cancer survival rates on account of advancements in cancer treatment and diagnostic procedures [4]. A 2003 comprehensive review of the literature indicated an incidence of multiple primary malignancies between 0.7% and 11.7% [5], while in 2017, another literature review reported an incidence between 2.4% and 17.2% [6]. The most common tumor location for multiple primary malignancies varies among the existing studies. In a US population study, a higher incidence of MPMs was found in patients with bladder (24%), skin (melanoma) (21.7%), kidney (21.6%), lung (20.5%), and colorectal cancer (19.8%) [7]. In Turkey, the most frequent tumor sites were the skin (17.1%), bladder (8.9%), and digestive system (small intestine, colorectum, anus) (7.7%), whereas, in China, MPMs were observed more frequently in patients with head and neck (5.65%) and urinary tumors (4.19%) [8,9]. Likewise, a number of different malignant tumor associations have been reported, including breast-breast, ovary–colon, ovary-breast [10], breast-uterine, head and neck-esophageal, head and neck-lung, head and neck-head and neck, prostate-bladder, prostate-colorectal, and colorectal-colorectal tumors [11].

The pathogenesis of multiple primary cancers is complex and not entirely understood. Cancer survivors may be prone to developing subsequent malignancies due to various intrinsic and extrinsic factors, including hormonal, lifestyle, environmental factors, genetic predisposition, and late carcinogenic effects of prior therapies.

Modifiable lifestyle factors, such as obesity, smoking, and alcohol consumption, are responsible for the great majority of cancer cases [12]. An increase in body mass index of 5 kg/m^2^ was strongly associated with esophageal adenocarcinoma and renal cancer in both men and women [13]. Cancer patients who had smoked tobacco regularly either before or at diagnosis have an increased risk of subsequent oral/pharyngeal, esophageal, stomach, lung, and hematological malignancies compared with never-smokers [14]. In regard to alcohol consumption, heavy drinkers present an increased risk of oral/pharyngeal, esophageal (squamous cell carcinoma), colorectal, laryngeal, and female breast cancer when compared with non-drinkers [15].

In certain individuals, the association of multiple primary tumors can be indicative of an underlying genetic susceptibility. Some well-known examples of hereditary cancer syndromes include Lynch syndrome (colon and endometrial cancer), von Hippel–Lindau disease (retinal and central nervous system hemangioblastomas, phaeochromocytomas, clear cell renal cell carcinomas, and pancreatic neuroendocrine tumors), multiple endocrine neoplasia type 2A (medullary thyroid cancer and pheochromocytoma), hereditary breast and ovarian cancer syndrome (breast and ovarian cancer), and Li-Fraumeni syndrome (sarcoma, breast cancer, brain tumors, leukemia, and adrenocortical carcinoma) [6].

In addition, cancer survivors may be prone to developing multiple primary malignancies due to the late carcinogenic effects of prior cancer therapies. Prostate cancer patients have a significant risk of subsequent bladder, colon, and rectal cancer if treated with external beam radiation therapy (EBRT) [16]. Survivors of Hodgkin’s lymphoma treated with chemotherapy (CHT) showed an elevated risk of secondary leukemia, non-Hodgkin’s lymphoma, pleural, liver, and lung cancer, whereas the combined treatment of EBRT and CHT showed an increased risk for developing leukemia, non-Hodgkin lymphoma, small intestine, bone, soft tissue, lung, and thyroid cancer [17].

Most MPMs presented and analyzed in clinical studies include dual primary malignancies. The association of three or more primary tumors is a rare occurrence and is thus often included in case reports and case series. In the literature, the reported frequency of triple primary malignancies ranges between 0.04% and 0.81% [18,19,20,21,22,23,24].

This research aims to determine the prevalence and clinicopathological characteristics of patients with triple primary malignancies, triple tumor association patterns, overall survival, and the correlation between survival time and independent factors.

## 2. Materials and Methods

This single-center retrospective study was conducted at the Institute of Oncology, “Prof. Dr. Ion Chiricuță”, Cluj-Napoca, a Romanian tertiary cancer center. The study population was identified through the institutional electronic medical records, and the study protocol was approved by the Ethics Committee of the Institute of Oncology, “Prof. Dr. I. Chiricuță”, Cluj-Napoca, approval number and date 234/18 February 2022. The study was conducted according to the principles of the Declaration of Helsinki, the International Conference on Harmonization Guideline on Good Clinical Practice, and Romanian laws and regulations. All patients signed the institutional informed consent for diagnostic and treatment procedures and for the use of their data in scientific reports, with personal data protection respected.

Over a twenty-five-year period, starting in 1996, 141,662 patients having at least one primary malignant tumor were admitted to our Oncology Institute. One hundred seventeen patients with triple primary malignancies, synchronous or metachronous, met the established eligibility criteria and were included in the study. The selection criteria included patients having three distinct histopathological malignant lesions, all confirmed by histopathological examination. Cases having MPMs that developed in the same organ or system were included in this study. We excluded the patients without histopathological confirmation of each tumor or with incomplete or ambiguous histopathological reports and in whom the second or third malignancy was suspected to be a metastasis, a recurrence, or an extension of previous tumors. In addition, patients with carcinoma of unknown primary origin were excluded from the analysis. Data such as gender, date of birth, time of diagnosis, site of origin, histology, survival from the first diagnosis, and date of death, if applicable, were gathered and recorded for each malignancy.

Tumors were organized into nine condensed groups as follows: head and neck (mouth and oropharynx, nasopharynx, larynx), digestive system (esophagus, stomach, small intestine, colorectum, liver, gallbladder), genital tract (cervix uteri, corpus uteri, ovary, vulva, prostate, testis), urinary tract (kidney, bladder), lung, skin, breast tissue, lymphoid and hematopoietic tissue, and others (thyroid gland, mediastinum, bone, joint, soft tissue, eye, brain, and central nervous system).

Every set of three primary tumors was sorted into groups based on the time of diagnosis as follows: (1) synchronous group (sMPMs), which includes three distinct primary tumors diagnosed within a six-month time frame, (2) metachronous group (mMPMs), which includes three distinct primary tumors diagnosed more than six months from each other, and (3) synchronous–metachronous group (smMPMs), which includes one metachronous and two synchronously occurring malignancies.

Collected data were statistically analyzed using the SPSS (version 25.0; SPSS Inc., Chicago, IL, USA) program. The Chi-squared test was used to evaluate intergroup differences, the Kaplan-Meier method was used to obtain the survival probabilities, and the Cox proportional hazard model was used to investigate the association between survival time and independent factors. All *p* values < 0.05 were considered statistically significant.

## 3. Results

### 3.1. Patient Characteristics

Out of 141,662 patients admitted to our Oncology Institute during the investigation period, 117 patients with triple primary malignancies, synchronous or metachronous, were included in the study. The observed prevalence of triple primary malignancies was 0.082%. Of these, 50.4% (*n* = 59) had one metachronous and two synchronous primary tumors (smMPMs), 32.4% (*n* = 38) had three metachronous tumors (mMPMs), and 17% (*n* = 20) had three synchronous tumors (sMPMs). In addition, 43.5% (*n* = 51) were male, and 56.5% (*n* = 66) were female, with a female-to-male ratio of 1.3:1. The median age at the first primary tumor diagnosis was 57. The median age observed in the sMPM, mMPM, and smMPM groups, respectively, was 70 (31–78), 49 (24–79), and 61 (17–79). During the observation period, 67.5% (*n* = 79) of patients included in the study had died, of which thirty-nine patients were female (49.3%), and forty (50.7%) were male (Table 1). At the first, second, and third tumor diagnosis, 73.5% (*n* = 86), 90.6% (*n* = 106), and 92.3% (*n* = 108) of patients were over the age of fifty (Table 2).

### 3.2. Time Interval between Tumor Diagnoses

The time interval between tumor diagnoses (time between the first and second, the second and third diagnosis) was evaluated for the synchronous–metachronous and the metachronous tumor groups. In the mMPMs group, the median time interval was 94.5 months between the first and the second tumor (ranging from 13 to 368 months) and 54 months between the second and the third tumor (ranging from 13 to 284 months). In the smMPMs group, the median time interval was six months between the first and the second tumor (ranging from 0 to 243 months), and eight months between the second and the third tumor (ranging from 0 to 165 months).

### 3.3. Distribution According to the Histological Type

Adenocarcinomas and squamous cell carcinomas account for 50% of all histological types identified. The most common histological types of the index tumor were squamous cell carcinoma (23%, *n* = 27), adenocarcinoma (21.4%, *n* = 25), and hematological malignancies (10%, *n* = 12). The most common histological types of the second primary tumor were adenocarcinomas (20.5%, *n* = 24), squamous cell carcinomas (17%, *n* = 20), followed by basal cell carcinomas and other specific carcinomas, which each accounted for 10.3% (*n* = 12). The most common histological types of the third primary tumor were adenocarcinoma (30.8%, *n* = 36), squamous cell carcinoma (16.2%, *n* = 19), and other specific carcinomas (15.4%, *n* = 18) (Table 3).

Among female patients, the majority of the histological types were adenocarcinomas (32.3%, *n* = 64), squamous cell carcinomas (15%, *n* = 30), and other specific carcinomas (14%, *n* = 28). Among male patients, the most frequent histological types were adenocarcinomas (30%, *n* = 46), followed by squamous cell carcinomas (24%, *n* = 36), and hematological malignancies (12.4%, *n* = 19) (Table 4).

### 3.4. Distribution According to the Tumor Site

Overall, the most common tumor sites included the genital organs (21%, *n* = 74), digestive organs (15.4%, *n* = 54), skin (14.8%, *n* = 52), and breast tissue (12%, *n* = 42). Among women, tumors were commonly found within the genital organs (27.7%, *n* = 55), breast tissue (20.2%, *n* = 40), and digestive system (12.6%, *n* = 25), while in men, they were mostly found within the digestive system (18.9%, *n* = 29), skin (18.3%, *n* = 28), and urinary tract (13.7%, *n* = 21) (Table 5).

The first primary tumor occurred mainly in the genital organs (28.2%, *n* = 33), breast tissue (15.4%, *n* = 18), and skin (12.8%, *n* = 15). The second primary tumor occurred in the genital organs (20.5%, *n* = 24), skin (19.7%, *n* = 23), and digestive organs (14.5%, *n* = 17). Most third primary tumors occurred in the digestive organs (22.2%, *n* = 26), genital organs (14.5%, *n* = 17), and skin (12%, *n* = 14) (Table 6).

When we look in more detail at the expanded groups, we notice that the most common tumor sites were the skin (14.8%, *n* = 52), breast tissue (12%, *n* = 42), colorectum (9.7%, *n* = 34), the lymphoid, hematopoietic tissue (8.8%, *n* = 31) and lungs (7.1%, *n* = 25). Among male patients, the most common tumor sites were the skin (18.3%, *n* = 28), lymphoid, hematopoietic tissue (12.4%, *n* = 19), colorectum (11.7%, *n* = 18), prostate (10.4%, *n* = 16), while among female patients, the most common tumor sites were the breast (20.2%, *n* = 40), skin (12%, *n* = 24), corpus uteri (11.6%, *n* = 23), colorectum (8%, *n* = 16) (Table 7).

In patients under the age of fifty, the most common tumor sites were the breast (17.8%, *n* = 5), cervix uteri (17.8%, *n* = 5), uterus (10.7%, *n* = 3), and skin (10.7%, *n* = 3) in women, and the hematopoietic, lymphoid tissue (21%, *n* = 4), colon (15.7%, *n* = 3) and testis (15.7%, *n* = 3) in men. In this age category, the most common associations were between genital tract tumors (cervical, uterine, ovarian), and breast cancer (26%, *n* = 5) in women, and hematopoietic, lymphoid tissue malignancies (lymphoma), and testicular cancer (33%, *n* = 3) in men.

In female patients, breast cancer was mainly associated with uterine (16%, n = 13), breast (15%, *n* = 12), skin (15%, *n* = 12), lung (10%, *n* = 8), colorectal (9%, *n* = 7), ovarian (7.5%, *n* = 6), and cervical cancer (7.5%, *n* = 6). When breast cancer is diagnosed as the index tumor, it is mostly associated with genital tract cancer (30.5%, *n* = 11) and contralateral breast cancer (13.8%, *n* = 5) as the second or third malignancy. The most frequently observed tumor associations in breast cancer patients were between breast–breast–uterine/cervical cancer (10%, *n* = 4). When genital tract cancer is diagnosed as the index tumor, it is commonly associated with genital tract (22%, *n* = 11), breast (14%, *n* = 7), and skin cancer (14%, *n* = 7) as the second or third malignancy. The most frequent tumor associations were between cervical–breast (18%, *n* = 6), uterine–breast (28%, *n* = 13), uterine–skin (17%, *n* = 8), ovarian–breast (25%, *n* = 6), and ovarian–uterine (16%, *n* = 4).

In male patients, prostate cancer was mostly associated with bladder (25%, *n* = 8), lung (21%, *n* = 7), hematopoietic, lymphoid tissue (15%, *n* = 5), and colorectal malignancies (12.5%, *n* = 4). When prostate cancer is diagnosed as the index tumor, it is predominantly associated with bladder (28.5%, *n* = 4), colorectal (14%, *n* = 2), lung (14%, *n* = 2), and kidney cancer (14%, *n* = 2) as the second or third malignancy. The most commonly observed associations in prostate cancer patients were between prostate–kidney–bladder cancer (12.5%, *n* = 2).

Lung cancer was commonly associated with breast (16%, *n* = 8), prostate (14%, *n* = 7), upper aerodigestive tract (14%, laryngeal *n* = 3, mouth and oropharyngeal *n* = 2, nasopharyngeal *n* = 1, esophageal *n* = 1), bladder (12%, *n* = 6), and gastrointestinal cancer (10%, *n* = 5). In female patients, lung cancer is associated with genital tract (31%, cervical *n* = 3, uterine *n* = 2, ovarian *n* = 2) and breast cancer (31%, *n* = 7), whereas in male patients, it is associated with prostate (25%, *n* = 7) and bladder cancer (21%, *n* = 6). In this group, the most frequently observed associations were between lung–bladder–prostate cancer (21%, *n* = 3) in men and lung–breast–uterine/cervical cancer (36%, *n* = 4) in women.

In patients with bladder cancer, the second and third malignancies were primarily prostate (26%, *n* = 8), kidney (20%, *n* = 6), and lung cancer (20%, *n* = 6). In this group, the most frequently observed associations were between bladder–breast–kidney cancer (66%, *n* = 2) in women and bladder–lung–prostate cancer (25%, *n* = 3) in men.

Kidney cancer was mostly associated with bladder (18%, *n* = 6), gastrointestinal (15%, *n* = 5), breast (15%, *n* = 5), hematopoietic, lymphoid tissue (12%, *n* = 4), and prostate cancer (9%, *n* = 3). The most commonly observed associations in kidney cancer patients were between kidney–bladder–prostate cancer (12.5%, *n* = 2).

Skin cancer was most frequently associated with skin cancer (32%, *n* = 34), hematopoietic, lymphoid tissue malignancies (14%, *n* = 15), and breast cancer (11.5%, *n* = 12). In female patients, skin cancer was associated with breast (25%, *n* = 12), uterine (16%, *n* = 8), and hematopoietic, lymphoid tissue malignancies (14.5%, *n* = 7), with the association of skin–breast–uterine cancer (12.5%, *n* = 3) being the most common. In male patients, skin cancer was primarily associated with other skin cancer lesions (50%, *n* = 28), and hematopoietic, lymphoid tissue malignancies (14%, *n* = 8), with the association of three skin malignancies being the most frequent (14.2%, *n* = 4).

An overall look at patients with colorectal cancer indicates that it is most commonly associated with colorectal (26%, *n* = 18), skin (11.7%, *n* = 8), breast (10%, *n* = 7), and prostate cancer (6%, *n* = 4). In women, colorectal cancer is associated with colorectal (37.5%, *n* = 12), uterine (15%, *n* = 5), and breast cancer (15%, *n* = 7), whereas in men, it is associated with colorectal (16%, *n* = 6), skin (16%, *n* = 6), prostate (11%, *n* = 4), and hematopoietic, lymphoid tissue malignancies (11%, *n* = 4).

When considering the nine condensed groups, the most commonly observed associations of three malignant tumors were between the genital tract–skin–breast tissue tumors (*n* = 5), skin–skin–skin tumors (*n* = 4), digestive system–genital tract–breast tissue tumors (*n* = 4), and genital tract–breast tissue–lung tumors (*n* = 4), accounting for 14.5% of all tumor associations (Figure 1).

### 3.5. Survival and Prognostic Factors

The median overall survival (OS) after the diagnosis of the first primary tumor was 148 months, with a 95% confidence interval (CI) (115–181). The median OS for the sMPMs group was 32 months, with a 95% CI (0–109), compared with 284 months, 95% CI (209–359) for the mMPMs group, and finally, 108 months, 95% CI (67–149) for the smMPMs group (Figure 2).

The 1-, 2-, 5- and 10-year survival rates for the mMPMs group were 0.97, 95% CI (0.92, 1.02), 0.97, 95% CI (0.92, 1.02), 0.95, 95% CI (0.88, 1.02) and 0.87 95% CI (0.76, 0.97). The 1-, 2- and 10-year survival rates for the sMPMs group were 0.70, 95% CI (0.5, 0.9), 0.59, 95% CI (0.37, 0.81), and 0.24, 95% CI (–0.11, 0.58). The 1-, 2-, 5- and 10-year survival rates for the smMPMs group were 0.95, 95% CI (0.89, 1.01), 0.92, 95% CI (0.84, 0.99), 0.76, 95% CI (0.65, 0.87) and 0.45, 95% CI (0.32, 0.58).

In order to avoid the accumulation of a type I error, we performed an overall comparison between the three MPMs group, which suggested that significant differences exist: chi-square(2) = 38.14, *p* < 0.001. Given that the overall comparison was statistically significant, we performed a pairwise comparison to detect the significant differences between specific MPMs group. We found significant differences regarding the median survival time between mMPM–sMPM groups, chi-square = 21.92, *p* < 0.001, and between mMPM–smMPM groups, chi-square = 31.06, *p* < 0.001. No significant differences were found regarding the median survival time between the sMPM–smMPM groups, chi-square = 2.99, *p* = 0.08.

The overall model containing the three MPM groups, age, and gender, was statistically significant: chi-square(4) = 47.31, *p* < 0.001. The male gender is associated with a higher mortality risk (0.56, 95% (0.35–0.90), *p* = 0.016) compared with the female gender. In addition, being over the age of fifty at the first tumor diagnosis increases the risk of mortality by 1.73, 95% (0.93, 3.22), but with no statistical significance (*p* = 0.086). In the sMPMs group, it was observed that the risk of mortality was 6.56 times higher 95% (2.74, 15.69), *p* < 0.001, than in the mMPMs group, whereas in the smMPMs group, the risk of mortality was 3.16 times higher, 95% (1.62, 6.19), *p* = 0.001, than in the mMPMs group.

## 4. Discussion

Cancer survivors may be prone to developing second and third primary malignancies due to various intrinsic and extrinsic factors, including hormonal, lifestyle, and environmental factors, genetic predisposition, and late carcinogenic effects of prior therapies [6]. The association of three or more primary tumors is a rare occurrence and is thus often included in case reports and case series. Among the existing data, the frequency of three primary tumors ranges between 0.04% and 0.81% [18,19,20,21,22,23,24]. In our study, the prevalence of triple primary malignancies was 0.08%, with a slight predominance of female patients (54.3%) and a female-to-male ratio of 1.3:1, as opposed to other reports on multiple primary malignancies where male patients were mostly predominant [18,25,26,27,28,29].

In agreement with the current literature [18,21,25,29], we also found that most patients (73%) were over the age of fifty at the first tumor diagnosis. When comparing the median age at the initial diagnosis, it was observed that regardless of gender, the lowest median age occurred in the metachronous group (*p* < 0.001), in agreement with a previous study stating that young people are more likely to develop metachronous tumors [26]. In addition, note that in female patients, the initial tumor occurred earlier, with over 31% being under the age of fifty.

Regarding gender distribution within the three groups, the majority of men were found in the smMPMs group, whereas the female patients were mainly distributed between the mMPM and smMPM groups. The synchronous–metachronous group was the most commonly found tumor setting, followed by the metachronous and the synchronous group. Similar results were presented in an epidemiological study analyzing 57 patients with at least three primary malignancies, where the “mixed form” (synchronous–metachronous) accounted for the majority of cases (58%), followed by the metachronous (33.3%) and the synchronous groups (8.8%). Gender distribution also showed that, among men, the “mixed form” is predominant (66.7%) [21].

According to GLOBOCAN 2020 [3], reports on cancer statistics revealed that the most prevalent cancers in Romania are colorectal, lung, and breast. Lung and prostate, followed closely by colorectal cancer, are most frequent among men, while breast, colorectal, and cervical cancer are most frequent among women.

We found that over half of the tumors were localized within the genital organs, digestive system, and skin. When analyzing the expanded groups, the most frequent tumor sites were the skin, breast tissue, and colorectum. Regardless of gender and consistent with previous findings [26], adenocarcinomas and squamous cell carcinomas account for most histological types. When looking into other reports analyzing patients with at least three primary malignancies, a study on 23 patients reported colorectum, lymphoid tissue, and prostate as the most prevalent tumor sites [29], while another reported colorectum, bladder, and prostate as the most frequent [21]. As opposed to our study, in both of these reports, the male gender was predominant, which may explain the differences.

Skin cancer (melanoma and non-melanoma) proved to be the most prevalent malignancy in our study (14.8%). The majority of skin cancer cases (71%) were diagnosed as the second or third primary malignancy and were mostly associated with other skin cancer lesions (32%). A higher occurrence of skin malignancies has been previously found in a study from a hospital-based cancer registry in Turkey, where the most common localization for multiple primary malignancies was also the skin (17.1%) [8]. It is believed that patients with basal cell carcinoma, squamous cell carcinoma, and melanoma have an increased risk of developing a second malignancy, especially skin melanoma in male patients [30]. Male patients with non–melanoma skin cancer have eight times the risk of later developing melanoma, while female patients have four times the risk compared with the general population [31]. These findings might be due to various factors, such as ultraviolet radiation exposure, phototype, personal or family history of skin cancer, type and degree of cumulative sun exposure, sun protection behavior [32], and possibly due to subsequent screening and complete clinical examination following the diagnosis of the index tumor.

Among female patients, the most common tumor sites for MPMs were the breast, skin, corpus uteri, and colorectum. Breast cancer was most frequently associated with uterine, breast, skin, lung, and colorectal cancer. In addition, 47% of patients with breast cancer as the initial tumor were later diagnosed with breast or genital tract tumors as the second or third malignancy. Women diagnosed with breast cancer present a greater risk of developing second gynecologic tumors [33], and according to other authors, there is also an increased risk of colorectal, lymphoma, melanoma, endometrium, and kidney cancers as second malignancies [34]. When evaluating the prevalence of primary malignancies occurring before breast cancer, malignant melanoma, gynecological malignancies, and gastrointestinal tumors were found to be the most common [35].

Skin malignancies have previously been reported to be associated with breast cancer [36,37,38]. A review of several epidemiological studies showed a significant association between cutaneous melanoma and breast carcinomas due to different epidemiological, genetic, and biological factors [39]. Concurrently, a study including 70,246 postmenopausal women found no significant association between non–melanomatous skin and breast cancer [40].

Among male patients, the most common tumor sites for MPMs were the skin, lymphoid and hematopoietic tissue, colorectum, and prostate. Patients with prostate cancer as their initial tumor were later diagnosed with bladder, lung, colorectal, and kidney cancer as the second or third malignancy. Prostate cancer survivors have an increased risk of subsequent primary malignancies of the lung, colorectum, and bladder [41]. Additionally, it has been shown that radiotherapy for prostate cancer was associated with a significant risk of later developing bladder, colon, and rectum cancer compared with unexposed patients [16].

Even though lung cancer is the leading cause of cancer death in our country [3], the number of triple primary malignancies involving the lungs was low (7.1%), and only 20% of lung cancer cases occurred as the initial diagnosis. It is worth considering that developing subsequent malignancies relies upon the prognosis and overall survival of the previous tumors. For this reason, patients with MPMs involving the lungs as the initial diagnosis have early-stage tumors compared with those having subsequent lung malignancies [42]. Lung cancer was most frequently associated with breast, prostate, bladder, gastrointestinal, and upper aerodigestive tract cancers.

In patients with bladder cancer, the second and third malignancies were primarily lung and prostate cancer. A 2016 study on second primary malignancies among survivors of common cancers revealed that bladder and lung cancer patients carry the highest risk of developing second primary malignancies. In addition, lung and prostate cancer were the most common subsequent malignancies among bladder cancer survivors [41]. These associations may underline common risk factors, such as tobacco consumption or exposure. It is known that tobacco products cause a variety of cancers, including lung, upper aerodigestive tract, digestive, bladder, and kidney [43]. Even though alcohol and tobacco use are a significant part of a patient’s history, part of the examined medical records lacked documentation regarding alcohol and tobacco use; therefore, we could not evaluate these common risk factors.

Colorectal cancer was most commonly associated with colorectal, skin, breast, and prostate cancer. Patients with colorectal cancer were most frequently diagnosed with a different colorectal cancer lesion as the second or third malignancy. This is in accordance with previous findings stating that patients with colorectal cancer demonstrate a higher risk of subsequent colorectal malignancies than the general population [44]. Additionally, pelvic radiation therapy has been associated with an increased rate of pelvic malignancies. Several studies demonstrated a greater risk of second primary rectal cancer in patients who received radiation therapy for prostate cancer [45,46,47]. Regarding the common association between breast and colorectal cancer, studies suggest that primary breast cancer does not increase the risk of subsequent colorectal malignancies [48,49].

Overall, the most common associations of triple primary malignancies include genital and breast tumors associated with either digestive, skin, or lung cancer, as well as the association of triple skin malignancies.

When considering the median overall survival, the mMPMs group showed a better median OS than the sMPM and smMPM groups. Patients with three synchronous tumors had a risk of mortality 6.5 times higher than those in the metachronous group, whereas patients with one metachronous and two synchronous tumors had a risk of mortality three times higher than those in the metachronous group.

Previous reports also revealed that tumor status was statistically significant in favor of metachronous tumors regarding OS [24,25,50]. We also observed that the male gender (0.56, *p* = 0.016), and being over the age of fifty at the first primary tumor diagnosis (1.73, *p* = 0.086) are associated with a higher risk of mortality in patients with triple primary malignancies. Our findings are in agreement with previous studies highlighting them as independent risk factors for mortality in patients with MPMs [26,51,52].

The association patterns of triple primary malignancies observed in the current study may underline common cumulative risk factors, such as environmental exposure to carcinogens and genetic susceptibility. Concurrently, the diagnosis and treatment of the first primary malignancy entail numerous tests and procedures over an extended period, along with regular screening for recurrence and subsequent malignancies, resulting in cancer survivors initially receiving more frequent screening for new primary breast, cervical, colorectal, and prostate cancer [53].

This study has several limitations. First, in order to address these rare events, we conducted a single-center retrospective study on a limited sample size, which can be subjected to various biases and may prevent our findings from being extrapolated. For these reasons, this study should not be used alone to draw strong conclusions about risk or causative factors. Second, this study was based on institutional electronic medical records that were not initially intended for research; therefore, variables, including family history, smoking status, alcohol consumption, dietary habits, and radiation exposure, were unavailable. Lastly, in our case, identifying triple tumor associations regarding the MPM groups (sMPMs, mMPMs, smMPMs) was found to be unfeasible.

## 5. Conclusions

Although our findings reinforce that triple primary malignancies are a rare occurrence, with an observed prevalence of 0.082%, the likelihood of a subsequent malignancy should always be considered throughout the short- and long-term surveillance of cancer patients to ensure prompt tumor diagnosis and treatment. The most common associations of triple primary malignancies include genital and breast tumors associated with either digestive, skin, or lung cancer, as well as the association of triple skin malignancies. Regarding tumor setting, the metachronous group showed better median overall survival than the sMPMs and smMPMs. Patients with three synchronous tumors demonstrated a risk of mortality 6.5 times higher than those in the metachronous group. In contrast, patients with one metachronous and two synchronous tumors showed a risk of mortality three times higher than those in the metachronous group. The male gender and being over the age of fifty at the first primary tumor diagnosis are also associated with a higher risk of mortality.

## Figures and Tables

**Figure 1 healthcare-11-00738-f001:**
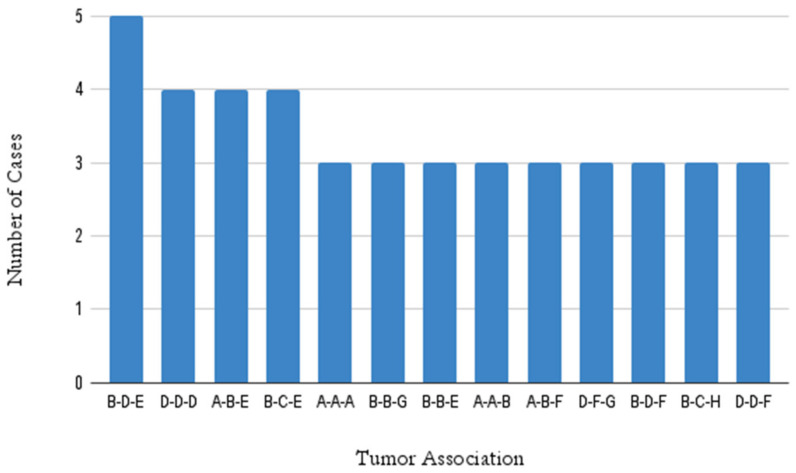
Distribution of triple tumors Association patterns. A—Digestive system; B—Genital tract; C—Lung; D–Skin; E—Breast tissue; F—Lymphoid and hematopoietic tissue; G—Other (includes the thyroid gland, mediastinum, bone, joint, soft tissue, eyes, brain, and central nervous system); H—Head and neck.

**Figure 2 healthcare-11-00738-f002:**
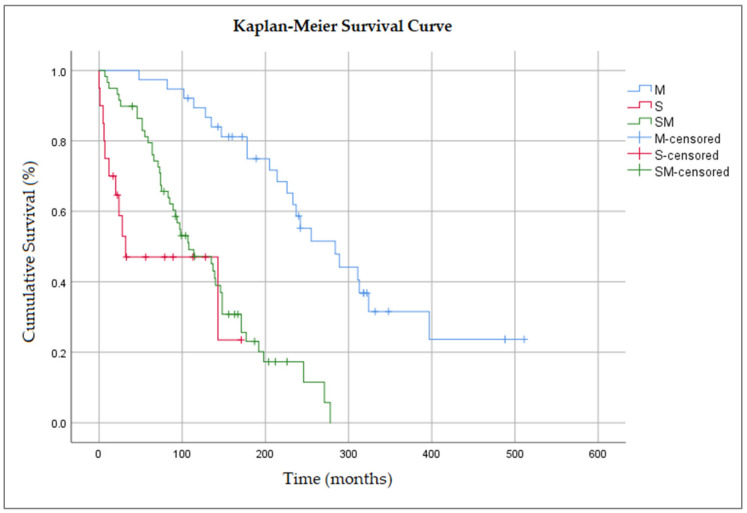
Kaplan–Meier survival curves comparing the overall survival in MPM tumor groups. M—Metachronous; S—Synchronous; SM—Synchronous–Metachronous.

**Table 1 healthcare-11-00738-t001:** Patient characteristics.

	sMPMs ^1^	mMPMs ^2^	smMPMs ^3^	Total	(*p*)
No. of Patients n (%)	20 (17)	38 (32.4)	59 (50.4)	117 (100)	
Gender Distribution n (%)					
Male Female	11 (9.4) 9 (7.6)	10 (8.5) 28 (24)	30 (25.6) 29 (24.7)	51 (43.5) 66 (56.5)	0.031
Median Age (years) at Index Tumor Diagnosis [range]					
Total Male Female	70 (31–78) 70 (31–78) 61 (43–78)	49 (24–79) 53 (35–79) 47 (24–70)	61 (17–79) 64 (17–79) 57 (31–78)	57 (17–79) 60 (17–79) 55 (24–78)	<0.001 0.025 <0.001
State at the End of the Observation Period–Deceased n (%)					
Total Male Female	11 (14) 6 (15) 5 (13)	22 (28) 7 (17.5) 15 (38)	46 (58) 27 (67.5) 19 (49)	79 (100) 40 (100) 39 (100)	0.051 0.039 0.639
State at the End of the Observation Period–Alive n (%)					
Total Male Female	9 (24) 5 (45) 4 (15)	16 (42) 3 (27) 13 (48)	13 (34) 3 (27) 10 (37)	38 (100) 11 (100) 27 (100)	

^1^ sMPMs—three synchronous primary tumors, ^2^ mMPMs—three metachronous primary tumors, ^3^ smMPMs—one metachronous and two synchronous primary tumors.

**Table 2 healthcare-11-00738-t002:** Age distribution based on gender and tumor groups.

	Age Years	Total	sMPMs ^1^ (n)		mMPMs ^2^ (n)		smMPMs ^3^ (n)		Overall ^4^ n (%)	
		n (%)	M	F	M	F	M	F	M	F
Age at 1st Diagnostic	<50	31 (26.5)	1	2	5	16	4	3	10 (19.6)	21 (31.8)
	>50	86 (73.5)	10	7	5	12	26	26	41 (80.4)	45 (68.2)
Age at 2nd Diagnostic	<50	11 (9.4)	1	2	1	2	3	2	5 (9.8)	6 (9.1)
	>50	106 (90.6)	10	7	9	26	27	27	46 (90.2)	60 (90.9)
Age at 3rd Diagnostic	<50	9 (7.7)	1	2	1	0	3	2	5 (9.8)	4 (6.1)
	>50	108 (92.3)	10	7	9	28	27	27	46 (90.2)	62 (93.9)

^1^ sMPMs—three synchronous primary tumors, ^2^ mMPMs—three metachronous primary tumors, ^3^ smMPMs—one metachronous and two synchronous primary tumors; ^4^ Overall—sMPM, mMPM, and smMPM groups; F—female, M—male; >50–age over fifty, <50—age under fifty.

**Table 3 healthcare-11-00738-t003:** Histological type distribution based on gender and the time of diagnosis.

Tumor Type	Total n (%)	First (n)		Second (n)		Third (n)		Overall ^4^ n (%)	
		M	F	M	F	M	F	M	F
Squamous Cell Carcinoma	66 (18.8)	13	14	11	9	12	7	36 (23.5)	30 (15.2)
Adenocarcinoma	85 (24.2)	15	10	15	9	16	20	46 (30.1)	39 (19.7)
Endometrioid adenocarcinoma	25 (7.1)	-	11	-	11	-	3	-	25 (12.6)
Transitional cell carcinoma	13 (3.7)	3	0	4	1	3	2	10 (6.5)	3 (1.5)
Basal cell carcinoma	18 (5.1)	3	1	4	8	1	1	8 (5.2)	10 (5.1)
Renal cell carcinoma	13 (3.7)	1	2	4	1	3	2	8 (5.2)	5 (2.5)
Invasive ductal carcinoma	19 (5.4)	1	7	0	7	0	4	1 (0.7)	18 (9.1)
Invasive lobular carcinoma	10 (2.8)	0	7	0	3	0	0	0 (0.0)	10 (5.1)
Melanoma	12 (3.4)	2	2	1	2	2	3	5 (3.3)	7 (3.5)
Bone and soft tissue sarcomas	6 (1.7)	2	0	0	1	1	2	3 (2.0)	3 (1.5)
Hematological Malignancies ^1^	31 (8.8)	7	5	8	3	4	4	19 (12.4)	12 (6.1)
Other specific Carcinomas ^2^	39 (11.1)	2	7	2	10	7	11	11 (7.2)	28 (14.1)
Other ^3^	14 (4.0)	2	0	2	1	2	7	6 (3.9)	8 (4.0)
Total	351 (100)	51	66	51	66	51	66	153 (100)	198 (100)

^1^ leukemia, Hodgkin and non-Hodgkin lymphoma, multiple myeloma; ^2^ undifferentiated carcinoma, papillary carcinoma, papillary microcarcinoma, neuroendocrine carcinoma, ductal carcinoma in situ, small cell carcinoma, Merkel cell carcinoma, mucinous carcinoma, ampullary carcinoma, medullary carcinoma; ^3^ seminoma, germinal cell tumor, nonseminomatous germ cell tumor, glioblastoma multiforme, anaplastic astrocytoma, neuroendocrine tumor, mesothelioma, mixed Müllerian tumor; ^4^ Overall—sMPMs, mMPMs, smMPMs; first—first diagnosed primary tumor; second—second diagnosed primary tumor; third—third diagnosed primary tumor; F—female, M—male.

**Table 4 healthcare-11-00738-t004:** Histological type distribution based on gender.

Histology Type	Total n (%)	sMPMs ^4^ (n)		mMPMs ^5^ (n)		smMPMs ^6^ (n)		Overall ^7^ n (%)	
		M	F	M	F	M	F	M	F
Squamous cell carcinoma	66 (18.8)	2	4	10	18	24	8	36 (23.5)	30 (15.2)
Adenocarcinoma	85 (24.2)	7	5	15	16	24	18	46 (30.1)	39 (19.7)
Endometrioid adenocarcinoma	25 (7.1)	0	3	0	7	0	15	0 (0.0%)	25 (12.6)
Transitional cell carcinoma	13 (3.7)	6	0	0	2	4	1	10 (6.5%)	3 (1.5)
Basal cell carcinoma	18 (5.1)	2	0	1	5	5	5	8 (5.2)	10 (5.1)
Renal cell carcinoma	13 (3.7)	5	1	0	2	3	2	8 (5.2)	5 (2.5)
Invasive ductal carcinoma	19 (5.4)	0	2	1	11	0	5	1 (0.7)	18 (9.1)
Invasive lobular carcinoma	10 (2.8)	0	0	0	4	0	6	0 (0.0)	10 (5.1)
Melanoma	12 (3.4)	2	1	0	2	3	4	5 (3.3)	7 (3.5)
Bone and soft tissue sarcomas	6 (1.7)	1	0	0	2	2	1	3 (2.0)	3 (1.5)
Hematological Malignancies ^1^	31 (8.8)	4	2	2	2	13	8	19 (12.4)	12 (6.1)
Other specific Carcinomas ^2^	39 (11.1)	2	7	0	11	9	10	11 (7.2)	28 (14.1)
Other ^3^	14 (4.0)	2	2	1	2	3	4	6 (3.9)	8 (4.0)
Total	351 (100)	33	27	30	84	90	87	153 (100)	198 (100)

^1^ leukemia, Hodgkin and non-Hodgkin lymphoma, and multiple myeloma; ^2^ undifferentiated car-cinoma, papillary carcinoma, papillary microcarcinoma, neuroendocrine carcinoma, ductal carci-noma in situ, small cell carcinoma, Merkel cell carcinoma, mucinous carcinoma, ampullary carci-noma, medullary carcinoma; ^3^ seminoma, germinal cell tumor, nonseminomatous germ cell tumor, glioblastoma multiforme, anaplastic astrocytoma, neuroendocrine tumor, mesothelioma, mixed Müllerian tumor; ^4^ sMPMs—three synchronous primary tumors; ^5^ mMPMs—three metachronous primary tumors; ^6^ smMPMs—one metachronous and two synchronous primary tumors; ^7^ Overall—sMPMs, mMPMs, smMPMs; F—female, M—male.

**Table 5 healthcare-11-00738-t005:** Tumor site distribution based on gender.

Tumor Site	Total n (%)	sMPMs ^3^ (n)		mMPMs ^4^ (n)		smMPMs ^5^ (n)		Overall ^6^ n (%)	
		M	F	M	F	M	F	M	F
Head and neck	19 (5.4)	0	2	5	3	7	2	12 (7.8)	7 (3.5)
Digestive organs	54 (15.4)	3	7	13	10	13	8	29 (18.9)	25 (12.6)
Lung	25 (7.1)	1	1	4	5	9	5	14 (9.1)	11 (5.5)
Genital organs	74 (21)	6	6	3	22	10	27	19 (12.4)	55 (27.7)
Urinary tract	31 (8.8)	12	1	1	6	8	3	21 (13.7)	10 (5.0)
Breast	42 (12.0)	1	5	0	21	1	14	2 (1.3)	40 (20.2)
Bone, soft tissue	5 (1.4)	1	0	0	1	2	1	3 (1.9)	2 (1.0)
Eye, brain, CNS ^1^	3 (0.9)	0	0	0	0	0	3	0 (0.0)	3 (1.5)
Skin	52 (14.8)	5	2	2	10	21	12	28 (18.3)	24 (12.1)
Lymphoid, hematopoietic tissue	31 (8.8)	4	2	2	2	13	8	19 (12.4)	12 (6.1)
Other ^2^	15 (4.3)	0	1	0	4	6	4	6 (3.9)	9 (4.5)
Total	351 (100)	33	27	33	81	93	84	153 (100)	198 (100)

^1^ CNS–central nervous system; ^2^ Other—thyroid gland and mediastinum; ^3^ sMPMs—three synchronous primary tumors; ^4^ mMPMs—three metachronous primary tumors; ^5^ smMPMs—one metachronous and two synchronous primary tumors; ^6^ Overall—sMPMs, mMPMs, smMPMs; F—female, M—male.

**Table 6 healthcare-11-00738-t006:** Tumor site distribution based on the time of tumor diagnosis.

Tumor Site	Total n (%)	sMPMs ^3^ (n)			mMPMs ^4^ (n)			smMPMs ^5^ (n)			Overall ^6^ n (%)		
		1st	2nd	3rd	1st	2nd	3rd	1st	2nd	3rd	1st	2nd	3rd
Head and neck	19 (5.4)	0	1	1	4	2	2	6	1	2	10 (8.5)	4 (3.4)	5 (4.3)
Digestive organs	54 (15.4)	4	2	4	4	7	12	3	8	10	11 (9.4)	17 (14.5)	26 (22.2)
Lung	25 (7.1)	0	1	1	2	2	5	3	4	7	5 (4.3)	7 (6.0)	13 (11.1)
Genital organs	74 (21)	4	5	3	12	5	8	17	14	6	33 (28.2)	24 (20.5)	17 (14.5)
Urinary tract	31 (8.8)	3	7	3	0	3	4	4	3	4	7 (5.9)	13 (11.1)	11 (9.4)
Breast	42 (12.0)	1	2	3	9	9	3	8	3	4	18 (15.4)	14 (12.0)	10 (8.5)
Bone, soft tissue	5 (1.4)	0	0	1	0	0	1	2	0	1	2 (1.7)	0 (0.0)	3 (2.6)
Eye, brain, CNS ^1^	3 (0.9)	0	0	0	0	0	0	0	1	2	0 (0.0)	1 (0.9)	2 (1.7)
Skin	52 (14.8)	3	1	3	5	7	0	7	15	11	15 (12.8)	23 (19.7)	14 (12.0)
Lymphoid, hematopoietic tissue	31 (8.8)	4	1	1	1	2	1	7	8	6	12 (10.3)	11 (9.4)	8 (6.8)
Other ^2^	15 (4.3)	1	0	0	1	1	2	2	2	6	4 (3.4)	3 (2.6)	8 (6.8)
Total (n)	351 (100)	20	20	20	38	38	38	59	59	59	117 (100)	117 (100)	117 (100)

^1^ CNS—Central nervous system; ^2^ Other—thyroid gland and mediastinum; ^3^ sMPMs—three synchronous primary tumors; ^4^ mMPMs—three metachronous primary tumors; ^5^ smMPMs—one metachronous and two synchronous primary tumors; ^6^ Overall—sMPMs, mMPMs, smMPMs; 1st—first diagnosed primary tumor; 2nd—second diagnosed primary tumor; 3rd—third diagnosed primary tumor.

**Table 7 healthcare-11-00738-t007:** Tumor site distribution based on gender–the expanded groups.

Tumor Site	n (%)	Male n (%)	Female n (%)
Head and Neck Mouth and oropharynx Larynx Nasopharynx Total	9 (2.6) 7 (2.0) 3 (0.9) 19 (5.4)	7 (4.5) 4 (2.6) 1 (0.6) 12 (7.8)	2 (1.0) 3 (1.5) 2 (1.0) 7 (3.5)
Digestive Organs			
Colorectum Small intestine Stomach Liver Gallbladder Esophagus Total	34 (9.7) 8 (2.3) 7 (2.0) 3 (0.9) 1 (0.3) 1 (0.3) 54 (15.4)	18 (11.7) 3 (1.9) 5 (3.2) 2 (1.3) 0 (0.0) 1 (0.6) 29 (18.9)	16 (8.0) 5 (2.5) 2 (1.0) 1 (0.5) 1 (0.5) 0 (0.0) 25 (12.6)
Lungs	25 (7.1)	14 (9.1)	11 (5.5)
Genital Organs			
Corpus uteri Cervix uteri Prostate Ovary Vulva Testis Total	23 (6.6) 16 (4.6) 16 (4.6) 12 (3.4) 4 (1.1) 3 (0.9) 74 (21)	– – 16 (10.4) – – 3 (1.9) 19 (12.4)	23 (11.6) 16 (8.0) – 12 (6.0) 4 (2.0) – 55 (27.7)
Urinary Tract			
Kidney Bladder Total	16 (4.6) 15 (4.3) 31 (8.8)	9 (5.8) 12 (7.8) 21 (33.7)	7 (3.5) 3 (1.5) 10 (5.0)
Breast	42 (12.0)	2 (1.3)	40 (20.2)
Bone, soft Tissue	5 (1.4)	3 (1.9)	2 (1.0)
Eye, brain, central nervous system	3 (0.9)	0 (0.0)	3 (1.5)
Skin	52 (14.8)	28 (18.3)	24 (12.1)
Lymphoid, hematopoietic tissue	31 (8.8)	19 (12.4)	12 (6.1)
Other			
Thyroid Mediastinum Total	14 (4.0) 1 (0.3) 15 (4.3)	5 (3.2) 1 (0.6) 6 (3.9)	9 (4.5) 0 (0.0) 9 (4.5)
Total (%)	351 (100)	153 (100)	198 (100)

## Data Availability

The data presented in this study are available on request from the corresponding author.

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
