# Peer review of "Triple Primary Malignancies: Tumor Associations, Survival, and Clinicopathological Analysis: A 25-Year Single-Institution Experience"

_healthcare, 2023, doi:10.3390/healthcare11050738_

Round 1

Reviewer 1 Report

The current manuscript is a long-term study of the clinical characterization of triple primary malignancies in a specific hospital setting. It is quite robust since data was collected over 25 years, the methodology in sound, and it is overall interesting to read. Nevertheless, some alterations should be made before acceptance for publication:

- In the introduction section specific information on the pathophysiology and prevalence of the types of cancer that were included in this study should be provided; an image regarding this topic could also be added, for better reader understanding;

- The limitations of this study should be further discussed, especially in what concerns sample size, and the possibility of drawing generalized conclusions for other hospitals, countries, etc., since this study was only conducted in one single institute.

Reviewer 2 Report

In the manuscript “Triple primary malignancies: tumor associations, survival, and clinicopathological analysis. A 25-year single‐institution experience”, the authors present and discuss about the differences in metachronous vs synchronous triple primary malignancies, as well as its related demographical data, mortality and most common tumor associations in different subgroups. There was a great effort to select the cohort.

The manuscript could be further improved by addressing the following points:

1)Line 32: Change first reference to “Billroth T. Allgemeine chirurgie pathologie UHD therapie. Reimer 1889: 908.”

2)Line 83: It would be interesting to sub-divide the urogenital group into uro and genital as they are very distinct from each other; as you’ve done in the first paragraph of section 3.4, table 5 and so on.

3)At section 3.3, it would be interesting to present the data for the first, second and third tumor for each gender, similar to what was shown in tables 4 and 6.

4)Line 166: bring “among women” to the beginning of the phrase. ‘Among women, tumors were…’

5)Line 197: what was the third most common tumor type in men in this cohort?

6)Line 220: “upper aerodigestive tract”. It would be interesting to sub-divide or further explain this, as done in lines 221&222 for genital tract.

7)Lines 234, 238 and 241: additionally to discussed risks, given the skin is our largest organ and all its malignancies have risk factors in common, it would be important to not consider skin cancer in more than one time point. Or keep the current analysis and compare with the proposed new one. Not much surprisingly, it is seen the results for men in line 241. Suggestion to add the results for men not considering skin in more than one timepoint.

8)Lines 242-249: Same as #7 applies to colorectal. Not much surprisingly, it is seen the results in lines 247-249. Suggestion to add the results not considering colorectal in more than one timepoint.

9)Lines 251-254: same as #2. Suggestion to keep the subdivision into uro and genital groups.

10)Based on #7 and #8, suggestion to adjust figure 1 accordingly or create another figure with the proposed new analysis.

11)Line 324: remove “one-“

12)Lines 400-402: same as #2. Suggestion to keep the subdivision into uro and genital groups.

13)Lines 433-435: same as #2. Suggestion to keep the subdivision into uro and genital groups.

Round 2

Reviewer 2 Report

Congratulations to the authors on this manuscript, thank you for such hard work. This is going to be an important addition to the literature.

Previous mentioned points were properly addressed by the authors.

Now that genital and urinary groups were subdivided, my only recommendation would be to double-check figure 1 if there is any tumor association involving the urinary tract that appeared at least 3 times that should be included in the figure. Given its not minimal prevalence in the cohort, for example higher than "lungs" group, which also appears at the figure, it was expected to see its association appearing with other tumors that share smoking as a common expressive risk factor.